# Novel Multifaceted Roles for RNF213 Protein

**DOI:** 10.3390/ijms23094492

**Published:** 2022-04-19

**Authors:** Giuliana Pollaci, Gemma Gorla, Antonella Potenza, Tatiana Carrozzini, Isabella Canavero, Anna Bersano, Laura Gatti

**Affiliations:** 1Laboratory of Neurobiology, Neurology IX Unit, Fondazione IRCCS Istituto Neurologico Carlo Besta, 20133 Milan, Italy; giuliana.pollaci@istituto-besta.it (G.P.); gemma.gorla@istituto-besta.it (G.G.); antonella.potenza@istituto-besta.it (A.P.); tatiana.carrozzini@istituto-besta.it (T.C.); 2Cerebrovascular Unit, Fondazione IRCCS Istituto Neurologico Carlo Besta, 20133 Milan, Italy; isabella.canavero@istituto-besta.it (I.C.); anna.bersano@istituto-besta.it (A.B.)

**Keywords:** RNF213, Moyamoya, arteriopathy, E3 ubiquitin ligase, angiogenesis, inflammation, PTP1b, lipid metabolism, antimicrobial activity

## Abstract

Ring Finger Protein 213 (RNF213), also known as Mysterin, is the major susceptibility factor for Moyamoya Arteriopathy (MA), a progressive cerebrovascular disorder that often leads to brain stroke in adults and children. Although several rare RNF213 polymorphisms have been reported, no major susceptibility variant has been identified to date in Caucasian patients, thus frustrating the attempts to identify putative therapeutic targets for MA treatment. For these reasons, the investigation of novel biochemical functions, substrates and unknown partners of RNF213 will help to unravel the pathogenic mechanisms of MA and will facilitate variant interpretations in a diagnostic context in the future. The aim of the present review is to discuss novel perspectives regarding emerging RNF213 roles in light of recent literature updates and dissect their relevance for understanding MA and for the design of future research studies. Since its identification, RNF213 involvement in angiogenesis and vasculogenesis has strengthened, together with its role in inflammatory signals and proliferation pathways. Most recent studies have been increasingly focused on its relevance in antimicrobial activity and lipid metabolism, highlighting new intriguing perspectives. The last area could suggest the main role of RNF213 in the proteasome pathway, thus reinforcing the hypotheses already previously formulated that depict the protein as an important regulator of the stability of client proteins involved in angiogenesis. We believe that the novel evidence reviewed here may contribute to untangling the complex and still obscure pathogenesis of MA that is reflected in the lack of therapies able to slow down or halt disease progression and severity.

## 1. Introduction

Ring Finger Protein 213 (RNF213), also known as Mysterin (Moyamoya steno-occlusive disease-associated AAA+ and RING finger protein), is a E3 ubiquitin protein ligase of 591 kDa consisting of 5207 amino acids encoded by locus 15,624 bp ORF and 5431 bp 5′ and 3′ UTRs on chromosome 17q25.3. The RNF213 gene is the major susceptibility factor for Moyamoya Arteriopathy (MA), a progressive cerebrovascular disorder that often leads to brain stroke in adults and children [1].

RNF213 is an atypical susceptibility gene, because its p.R4810K variant has been associated with MA mainly in East Asian patients [2]. Moreover, the pattern of inheritance in MA familial cases is unclear and most likely heterogeneous. Although several rare and nondisruptive RNF213 variants, distinct from p.R4810K, have been reported in MA patients of European ancestry, no major susceptibility variant has been identified to date in Caucasian MA patients [3]. In addition, the RNF213 variants that have been reported so far in Caucasian cases did not lead to conclusions useful for identifying putative therapeutic targets for MA patient treatments. The difficulty in drawing a clear conclusion about causality when RNF213 variants were considered individually in patient cohorts and the low penetrance observed in most families has rendered genetic counseling quite difficult so far. For these reasons, the investigation of novel biochemical functions, substrates and unknown partners of RNF213 will help to unravel the pathogenic mechanisms of MA and will facilitate variant interpretations in a diagnostic context in the future. The incomplete disease knowledge on pathogenic drivers is reflected in the lack of therapies able to slow or halt the disease progression and severity. The aim of the present review is to discuss novel perspectives regarding emerging RNF213 roles in light of the recent literature updates and dissect their relevance for understanding MA and for the design of future research studies.

## 2. RNF213 from Gene to Molecular Structure

The RNF213 gene is conserved in protochordates and vertebrates, and it is ubiquitously expressed, with the higher expression rate in immune tissue [2,4]. The encoded protein exists in two isoforms [4].

RNF213 is a cytosolic protein with a perinuclear space localization. It is a huge unique protein that encompasses both ubiquitin ligase activity and ATPase activity; its functional domains are divided as follows:

−Six ATPase domains, each containing a Walker A and a Walker B domain, peculiar to ATP-binding proteins; the A motif is the main ‘P-loop’ responsible for binding phosphate, while the B motif is a magnesium-binding motif [5].−One RING finger domain with an E3 ubiquitin ligase domain that is able to covalently modify the substrate with the addition of ubiquitin molecules and stimulate the intracellular biological processes, such as proteasomal protein degradation [4].

Morito and colleagues identified the presence of two tandem ATPase domains and demonstrated that RNF213 is capable of forming a ring-shaped homo-oligomer in the cell approximately comparable with macromolecular complexes such as the ribosome. They also observed that a relatively large amount of protein diffuses as a monomer in the cell, suggesting that Mysterin exists in a balanced condition between the monomeric and oligomeric states [6].

A recent cryo-EM analysis by Ahel and colleagues showed a detailed structure of the full-length murin RNF213 (584 kDa, 5184 residues), possibly resembling the human homolog protein [1]. The murine protein is depicted by 20 subdomains divided into three structural components: the N-arm, AAA and E3 module. The first one (residues 1–1290) is connected by a linker domain (1291–1774) to the second module, the AAA core, made up of six AAA units (1775–3405). The central region consists of a hinge domain (3406–3588) that connects the AAA core to the E3 domain (3598–4926). The last is composed of a four-domain scaffold that locates the E3 RING (3940–3999) at the edge of the molecule at the opposite site of the AAA domain. They demonstrated the presence of a six-membered AAA ring, with two catalytically competent units (AAA3 and AAA4), bearing all functional motifs (Walker A and Walker B) [1].

The RING finger domain is composed of a E3 module with ubiquitinase activity. Post-translational protein modifications by mono- or polyubiquitin chains are involved in several signaling pathways and are tightly regulated to ensure the cellular processes. Ubiquitin molecules are conjugated at the ε-amino group of lysyl residues of the target proteins through isopeptide bonds. The nature of the ubiquitin–ubiquitin isopeptide bond appears to determine the subsequent fate of ubiquitinated proteins [7].

Several studies have proposed that Mysterin exerts ubiquitylation activity toward a variety of substrate proteins, including itself, with a mechanism of autoubiquitylation [4,8]. The ubiquitin ligase activity of RNF213 is cysteine-dependent, capable of promoting the ubiquitin transfer by a trans-thiolation mechanism rather than by activating the E2–Ub conjugate, as Ubiquitin E3 ligases conventionally do. RNF213 employs a E3 scaffold to perform its ubiquitin ligase function in a RING-independent manner [1]. To date, Mysterin is the only known protein that exerts both AAA+ ATPase and ubiquitin ligase activities, but how it coordinates the unique combination of enzymatic activities to perform specific functions in a cell remains elusive.

## 3. RNF213 Genetic Background

Despite the growing and detailed information that recently was brought to light about its structure, the exact function of this huge protein still remains unclear. Since RNF213 has a critical role in MA, its correlation in vascular development/angiogenesis was the first to be defined. In a locus-specific GWAS, RNF213 was identified for the first time as a susceptibility gene for MA. Due to the unusually high prevalence of MA patients in Japan (6 per 100,000 population), the presence of a founder mutation among the Japanese population was hypothesized. A GWA study demonstrated that 19 of the 20 MA families shared the same single-base substitution in exon 60 of RNF213: c.14576G>A, causing an amino acid substitution from Arg to Lys in the 4810 position (R4810K). It was identified in 95% of MA familial cases, 73% of non-familial MA cases, including 45 heterozygotes and a single homozygote, and 1.4% of controls. Additional missense mutations were detected in three non-familial MA cases without the R4810K mutation, suggesting that such mutations greatly increase the risk of MA, with an odd ratio (OR) of 190.8 [2]. Studies based on a large-scale sequencing analysis tried to extend the correlation to all East Asian populations, identifying this genetic variant as common in 42 families [4,9].

A different genetic background in Caucasian patients was found through the positive association of MA with rare heterozygous RNF213 missense variants, particularly in early onset and/or familial cases. These ‘Caucasian’ variants significantly clustered in a C-terminal ‘hotspot’ encompassing the RING finger domain [3]. RNF213 rare pathogenic missense variants were also detected in all affected members of some MA European families also carrying mutations in PALD 1 (Phosphatase Domain Containing Paladin 1), a gene probably involved in peptidyl-tyrosine dephosphorylation. Nonetheless, three carriers were unaffected, and one of them presented a steno-occlusive angiopathy without fulfilling the criteria for MA, similar to the known incomplete penetrance of RNF213 variants [10]. In addition, a WES study on European MA led to the identification of de novo germline heterozygous CBL mutations in unrelated cases, presenting a bilateral severe early onset MA. Likewise, RNF213, the CBL gene, encodes a huge E3 ubiquitin protein ligase containing a RING finger domain and is involved in proteasome modulation. This finding strongly reinforces and supports the involvement of a defective proteasome signaling pathway in MA pathophysiology [11].

## 4. RNF213 and Inflammation-Related Angiogenesis

Kobayashi and colleagues in 2015 obtained biochemical and functional characterizations of the RNF213 R4180K mutation in angiogenesis through in vitro and in vivo studies. They showed that the upregulation of RNF213 could be produced by inflammatory signals. IFN-B was able to increase RNF213 gene expression through the STAT-x-binding site in the promoter region. The upregulation mediated by IFN-B was associated with lower angiogenesis, and RNF213 R4810K polymorphism led to a decreased tube-forming ability in response to environmental stimuli [12].

Accordingly, Ohkubo and colleagues demonstrated that RNF213 is transcriptionally induced by the administration of TNF-A and co-stimulation with other proinflammatory cytokines, thus connecting the environmental factors to the RNF213 response. RNF213 was also involved in the cell proliferation of endothelial cells through decreasing AKT phosphorylation and inducing matrix metalloproteinase-1 (MMP1) [13]. These findings support the importance of the inflammatory environment in MA. In particular, MA-involved vessels are depicted by a concentric fibrocellular hyperplasia of the intima [14,15], caused by the proliferation of vascular smooth muscle cells (VSMC) and extracellular matrix (ECM), resulting in a progressive intimal fibrous thickening [16,17,18]. Since little is known about fibrosis in MA, the cross-talk between endothelial cells (EC) and VSMC in MA is under investigation. Some evidence suggested that chemokines derived from defective colony-forming EC could be responsible for the aberrant recruitment and proliferation of VSMC progenitors in MA patients [19]. Recently, it has been demonstrated that ECM receptor-related genes are significantly downregulated in iPS-derived ECs from patients carrying the p.R4810K mutation [20]. The latter cells produced less ECM components than the controls, suggesting that RNF213 variants may be directly involved in changes in the ECM [21]. Thus, it is likely that VSMC are implicated in fibrosis and EC in the fibrillogenesis process through chemokine secretion in an inflammatory context.

Ubc13/Uev1A (Ubiquitin-conjugating enzyme) is an E2 enzyme that cooperates with the RNF213 RING domain, promoting K63-linked polyubiquitination but not K48-linked polyubiquitination of the substrates [22]. K48 ubiquitination implies subsequent proteolytic degradation of the target protein [23]. Conversely, K63 linkages are known to regulate critical cellular processes such as DNA repair, innate immune response, the clearance of damaged mitochondria and protein sorting [24]. Moreover, K63 ubiquitination plays a pivotal role in the regulation of both canonical and noncanonical NF-κB activation pathways that negatively regulate apoptosis. The Ubc13/RNF213 interaction may contribute to the selective ubiquitination of the target protein through a mechanism in which Ubc13 forms heterodimers with other E2 enzymes and catalyzes the synthesis of K63-polyubiquitin chains [22].

Ahel et al. evaluated the R4753K mutation in the murine protein homologous to RNF213 R4810K and demonstrated that it does not significantly alter the enzymatic properties of RNF213 in vitro. MA mutations strongly cluster and alter the overall conformation and dynamics of the composite E3 module and, thus, its ubiquitination activity [1].

The R4810K variant undoubtedly afflicts the ubiquitin ligase domain of RNF213, leading to either mechanistic inhibition or altered substrate binding.

Bahn and colleagues in 2016 proposed RNF213 as a promoter of angiogenesis in an in vitro tumor model through the stabilization of the master angiogenesis regulator HIF-1 (hypoxia inducible factor 1). They demonstrated for the first time that RNF213 is a putative substrate of PTP1b, and it is a major regulator of the ubiquityloma—in particular, in HER2+BC cells [25]. They implicated RNF213 in vascular development through its direct/indirect effect in the angiogenic pathway, thus conferring it a leading role in the pathophysiology of MA.

Inflammation is extensively studied in correlation with obesity and metabolic disorders. In the presence of obesity, the recruitment of adipokines and chemokines causes the infiltration of macrophages into the adipose tissues, where they secrete proinflammatory molecules that lead to insulin resistance [26]. Inflammation triggers the induction of TNFα in adipocytes, which further induces the expression of PTP1B [27]. Interestingly, the RNF213 interactome included a cluster belonging to TNFα and PTP1B [8]. PTP1B is also a negative regulator of insulin [28], while TNFα causes insulin resistance [29] and acts as an antiadipogenic factor by altering PTP1B [28]. PPARγ has been suggested to be one of the main adipogenic regulators [8]. To identify a possible correlation between TNFα/PTP1B and RNF213 in adipogenesis, a PPARγ activator and anti-inflammatory molecule was used [30]. The insulin-resistant TNFα/PTP1B pathway enhances RNF213 expression, while PPARγ-mediated insulin sensitization suppresses its expression. Thus, it was hypothesized that PTP1B could inactivate PPARγ to induce RNF213 [8]. The RNF213 expression appears linked to inflammation and the insulin pathway. Probably, TNFα induces the expression of PTP1B, increasing its activity [31,32], which blocks PPARγ [27]. PPARγ can suppress the transcription of RNF213; therefore, the suppression of PPARγ induces RNF213. RNF213 emerged as a link between obesity, inflammation and insulin resistance.

## 5. RNF213 Key Interactors

The ‘mystery’ of RNF213 also wraps around its interactors. As mentioned above, Banh and colleagues, studying the tumor hypoxia microenvironment, stumbled onto RNF213. They investigated the factors involved in nonmitochondrial oxygen consumption (NMOC), a mechanism aimed at limiting oxygen consumption and promoting tumor survival in the hypoxia microenvironment. PTP1B is required for Her2/Neu-driven breast cancer in mice. PTP1B deficiency sensitizes HER2^+^ breast cancer cell lines to hypoxia by increasing NMOC by α-KG-dependent dioxygenases (α-KGDDs). α-KGDDs probably acts by catalyzing the prolyl hydroxylation of HIF1α and determining its destabilization and the consequent potential failure of the hypoxic response [33]. PTP1B probably acts via RNF213 to suppress α-KGDD activity and NMOC. It was suggested that PTP1B negatively regulated RNF213 E3 ligase activity and that some RNF213 substrates globally regulate α-KGDDs by altering the level of co-substrates/metabolites, such as ascorbate and/or iron [25]. It could be possible that this pathway, allowing tumor growth in the hypoxia microenvironment, can be compromised in MA patients.

Recently, a new antimicrobial activity of RNF213 as an interferon-induced ISG15 interactor/ISG15-binding protein and cellular sensor of ISGylated proteins was discovered (see below, Section 7) [34]. ISG15 was reported to interact in a noncovalent manner with HIF1α, preventing its dimerization and downstream signaling [35]. The PTP1B-negative regulation of RNF213 was associated with the mechanism involving the IFN-I-induced oligomerization of RNF213. It would be worthwhile to test whether PTPB1 also regulates RNF213 oligomerization in response to interferon, since PTP1B also affects JAK-STAT signaling [34].

## 6. RNF213 and Lipid Metabolism

The lipid accumulation interferes with normal cellular and tissue functions, causing lipotoxicity, which seems to be one of the causes of many metabolic diseases [36,37]. Saturated fatty acids are implicated in several mechanisms, including incorporation into membrane lipids and storage in lipid droplets, by acting as protein modifiers or mitochondrial oxidation. Palmitate (C16: 0) is particularly toxic for cells, since high concentrations of palmitate induce apoptosis [38]. More recently, Piccolis et al., conducted studies on palmitate-induced toxicity. An excess of palmitate is incorporated into a wide variety of lipids, and the accumulation of saturated glycerolipids in the ER triggers the prolonged activation of the IRE1 pathway of the UPR [39,40]. The depletion of RNF213 was identified as a protection mechanism against palmitate-induced lipotoxicity, reducing the cellular toxicity by 50%. Furthermore, RNF213 knockdown also lessened palmitate-induced cell death in different cell types. The most surprising finding is that the depletion of RNF213 almost normalized the cellular lipidome during exposure to palmitate, abrogating the accumulation of di-saturated glycerophospholipids. Measuring the incorporation of radiolabeled palmitate into lipids, it seems that RNF213 knockdown did not alter the total palmitate incorporation but promoted the partitioning of the substrate toward triglycerides. Although the mechanism remains to be determined, RNF213 is clearly a modulator of lipotoxicity from saturated fatty acids through a mechanism that has an important effect on a cell’s ability to store lipids in droplets, ubiquitous organelles specialized for neutral lipid storage (LDs) (Figure 1).

The first evidence that RNF213 is targeted at LDs was reported by Sugihara and colleagues [41]. To dissect the subcellular distribution of RNF213, they performed high-resolution fluorescence microscopy with RNF213 harboring mCherry (mCherry-mst). They found that cells transiently overexpressing mCherry-mst formed LDs more extensively than intact (non-transfected) cells. A quantitative analysis of those LDs revealed that the number of LDs and the area occupied by LDs are markedly increased in RNF213-overexpressing cells. Conversely, the depletion of RNF213 by CRISPR/Cas9 or siRNA resulted in a significant reduction of LDs. The authors reported that RNF213 is targeted at LDs and markedly increases their abundance in cells. This effect was exerted primarily through specific elimination of adipose triglyceride lipase (ATGL) from LDs. However, no physical interaction between RNF213 and ATGL was found. A plausible scenario is that RNF213 and ATGL compete for binding to a common anchoring protein facilitating LD localization. RNF213 may affect the putative anchoring protein with its AAA+ activity and prevent the ATGL influx to LDs [41] (Figure 2).

Previous epidemiological studies did not find any correlation between MA and obesity/dyslipidemia, as the condition is characterized by the hyperplasia of vascular smooth muscle cells (VSMC) and by luminal thrombosis to the injury, while, generally, no atherosclerotic changes were found [42]. Therefore, MA was not considered closely related to the lipid metabolism. However, very recently, it was suggested that lipid metabolism may be involved in the pathogenesis of MA, although neither study showed a direct association between the RNF213 mutations and dyslipidemia [43]. In fact, the untargeted gas chromatography mass spectrometry (MS) approach identified 25 discriminating serum metabolic biomarkers in MA patients. A panel of fatty acids (myristic acid, pelargonic acid, palmitic acid, palmitoleic acid and stearic acid) could be used to distinguish between MA patients and healthy donors (HD). The level of succinic acid, an intermediate of the TCA cycle, was significantly decreased in the MA patient serum, indicating an alteration in the cycle that would lead to an altered level in the fatty acid. A defective respiratory machinery in the mitochondria could be associated with the mitochondrial abnormalities observed in MA patients [44,45], and with a decrease of the ‘building blocks’ of cellular membrane and defective signal transduction [46].

An untargeted lipidomic approach was recently performed on the plasma of MA patients [47], confirming what was previously observed in the serum [46]. Indeed, MA patients showed a surprisingly lower plasma content of lipids as compared to HD, particularly in lipids belonging to the glycosphingolipid and phospholipid classes. Through a discriminant analysis (PLS-DA), the lipidomic profile showed a separation of 23.1% of the principal component (PC1). The authors supposed that, since glycosphingolipids and phospholipids are plasma-membrane components, their reduced level could correlate with a decrease of the cellular debris due to a reduction of peripherally circulating progenitor cells [48] as an effect of the major cerebral recruitment of circulating cells [49]. An increase of cardiolipin, a typical mitochondrial lipid, could confirm the frequently reported abnormalities of MA patients. A quantitative targeted MS analysis showed a free spingoid base level increase in the plasma of MA patients in comparison to HD. Specifically, Sph, DHSph, S1P and DHS1P concentrations were augmented in MA patients. Of note, S1P and DHS1P are pro-angiogenic/proliferative factors, again in agreement with MA pathogenesis. All studies focused on novel unreleased and unexpected RNF213 roles in lipid metabolism, thus widening the scenario of the putative functions of Mysterin.

## 7. RNF213 and Antimicrobial Activity

The RNF213 gene may also impact the resistance to infectious diseases such as Rift Valley Fever (RVF), an emerging viral zoonosis affecting ruminants and humans [50]. Evidence from experimental models has demonstrated the importance of genetic host factors in determining the RVF severity in mice. The authors narrowed down the critical interval to a 530 kb region containing five protein-coding genes, among which RNF213 emerged as a potential candidate. They generated Rnf213-deficient mice by CRISPR/CAS9 and showed that they were significantly more susceptible to RVF than the control mice. The STRING Mus musculus database analysis revealed that murine RNF213 formed a highly interconnected network with UBA7, USP18, PARP14, IFIT3, IRF7, RSAD2, STAT1, IGTP, MX2, RTP4, OASL2, IRGM2, LGALS3BP and OAS2 proteins. Seven of these proteins were associated with the Gene Ontology (GO) categories ’innate immune response’, six with the ’immune effector process’ and/or ’response to other organism’, five with ’defense response to virus’, three with ’negative regulation of viral process’ and two with ’response to type I interferon’. These numerous interactions with innate immunity genes suggested a role of RNF213 in response to infection [50]. Homozygous Rnf213-deficient mice (referred to as Rnf213tm3/tm3) did not show any visible phenotype under a conventional environment. After intraperitoneal injection of the RVF virus (RVFV), the survival time in lethally infected mice was significantly shorter in Rnf213tm3/tm3 than in Rnf213+/+. These results indicated that RNF213 gene delays the fatal outcome of RVFV infection. In humans, the primary site of RVFV replication is in the liver, together with LD accumulation [51]. A major consequence of RVFV infection in mice is the overwhelming infection of hepatocytes resulting in early-onset death for severe hepatitis [52,53]. It has been observed that the hepatocytes store up cytoplasmic LDs persisting during liver regeneration in the surviving mice [54]. It is therefore possible that RNF213 influences the resistance to RVFV infection through a major role in lipid metabolism [50] (Figure 3).

An in silico assessment of the RNF213 expression profile in a large collection of human tissues confirmed the high correlation between the RNF213 and GO categories of ‘immune response to virus’ [13]. A few years later, emerging data pointed out the identification of RNF213 as a novel immune sensor, unveiling the link between MA and infection [55]. In particular, the ubiquitin coat, which marks cytosol-invading bacteria as the cargo for antibacterial autophagy [56,57,58], is likely to be the result of the unprecedented ubiquitylation of a minimal substrate, such as the lipid A of bacterial lipopolysaccharide (LPS), mediated by the RNF213 ubiquitin ligase [55]. This intriguing result has been obtained by the RNAi approach and CRISPR technology used to test whether RNF213 is required for the ubiquitylation of LPS. Indeed, cells lacking RNF213 failed to ubiquilate LPS upon infection. In particular, this study demonstrated that RNF213-mediated LPS ubiquitylation requires a catalytically active AAA+ module, independent of the RING domain. Through the creation of a bacterial ubiquitin coat, RNF213 was able to restrict the proliferation of cytosolic Salmonella, suggesting an extension of the scope of ubiquitylation more than a simple and basic post-translational protein modification [55] (Figure 4). Overall, bacterial/viral infections could contribute to MA development in genetically susceptible subjects, although no RNF213 variant predisposed to MA was impaired during the LPS ubiquitylation ability [55].

RNF213 has been described as an ISG15 interactor and cellular sensor of ISGylated proteins [34]. ISG15 is an interferon-stimulated and Ub-like protein that, in a process known as ISGylation, conjugates an immunity-related modification, counteracting microbial infection [59,60,61,62,63]. Once upregulated by type I and III interferons, viral nucleic acids, bacterial DNA and LPS, ISG15 is able to exert potent antiviral effects both in vitro and in vivo [64,65,66]. It has also been described as an ISG15 antibacterial activity against intracellular bacterial and eukaryotic pathogens [64,67,68]. Currently, the mechanisms by which ISG15 modification is perceived and how it protects against microbial infections have not yet been clarified. A supposed model is based on the localization of the ISG15 E3 ligase (HERC5) at the ribosome site where the proteins are co-translationally modified by ISG15 during infection, thus hampering with the function of newly translated viral proteins [69]. Differently from ubiquitin, ISG15 exerts an antimicrobial role by also conjugating to target substrates through noncovalent interactions or acting as a secreted cytokine. For instance, free ISG15 is referred to interact in a noncovalent way with HIF1α, avoiding its dimerization and downstream signaling [35]. For this reason, in order to identify and map the noncovalent ISG15 protein–protein interactions, Thery et al. developed the MS-based approach named Virotrap, a virus-like particle trapping technology that allows the capture of protein complexes within virus-like particles (VLPs) budding from mammalian cells. Thanks to this technology, they mapped the noncovalent interactome of ISG15 in human cells. RNF213 has been identified as the most enriched protein in the ISG15 VLPs, requiring both the N- and C-terminal domains to selectively interact with ISG15. This evidence highlighted the capability of RNF213 to interact with and bind ISGylated proteins, recruiting them to LDs. Monomeric RNF213 is directly recruited from the cytosol on the surface of LDs, where its oligomerization in a hexamer is driven by ATP binding. RNF213 oligomerization and translocation to LDs may serve for the creation of a binding platform for ISG15 and, potentially, multiple ISGylated proteins [34]. Thery et al. demonstrated that the increasing expression of RNF213 is strictly associated with lower viral infection levels in cultured cells. Conversely, the reduced RNF213 expression levels promoted in vitro infection with different viruses. Compared to the antiviral activity, the antimicrobial effect of RNF213 was more marked and subjected to ISG15. Taken together, these results emphasized the role of RNF213 in the innate cellular immune response and pointed out a functional association between ISG15 and RNF213. As far as the antimicrobial activity reported in vivo, it is likely that the overexpression of RNF213 prior to infection gives rise to the innate defense response (Figure 5).

## 8. Conclusions

Our literature review resumes and summarizes the multiplicity of the RNF213 roles, including novel unexpected functions. The RNF213 gene was first discovered in 2011, and since then, the scientific setting has been filled with subsequent in vitro and in vivo studies, expanding the knowledge of such a mysterious and peculiar protein. Since its identification as a susceptibility gene in MA, its involvement in angiogenesis and vasculogenesis has strengthened together with a role in inflammatory signals and proliferation pathways. Several previous studies have shown that RNF213 may also be involved in other vascular phenotypes, such as premature coronary artery disease, renal or aortic artery disease [70], hypertension [71], cerebral cavernous malformation [72], fibromuscular dysplasia [73], stenosis/major intracranial artery occlusion [74] and intracranial aneurysm [75]. Therefore, it is clear that RNF213 has a relevant role in many vascular diseases, although the specific molecular mechanisms in which it is involved have not always been clarified. Interestingly, most recent studies have been more focused on its relevance in antimicrobial activity and lipid metabolism, highlighting new, intriguing perspectives. This novel research field may corroborate a major role of RNF213 protein in the proteasome pathway by representing the protein as an important regulator of the stability of the client proteins involved in angiogenesis. (i.e., HIF1α [25]). All these studies underlined that RNF213 acts as an antimicrobial host defense effector in MA patients, possibly triggered by autoimmune responses, prior infection and/or inflammatory processes. It is reasonable to speculate that MA patients carrying RNF213 polymorphisms may exhibit a defective immune response to infections or may be more predisposed to autoimmune reactions. Therefore, autoimmune diseases and infections, which are MA-associated conditions, could trigger the disease in genetically susceptible individuals [76].

We believe that the novel evidence reviewed here may contribute to untangling the complex and still obscure pathogenesis of MA that is reflected in the lack of therapies able to slow or halt disease progression and severity.

## Figures and Tables

**Figure 1 ijms-23-04492-f001:**
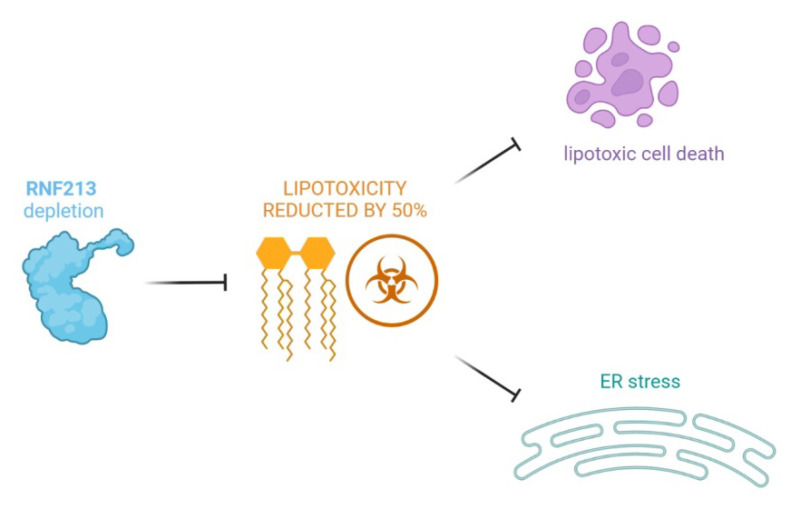
A protective mechanism mediated by the RNF213 protein against palmitate-induced lipotoxicity (ER, endoplasmic reticulum).

**Figure 2 ijms-23-04492-f002:**
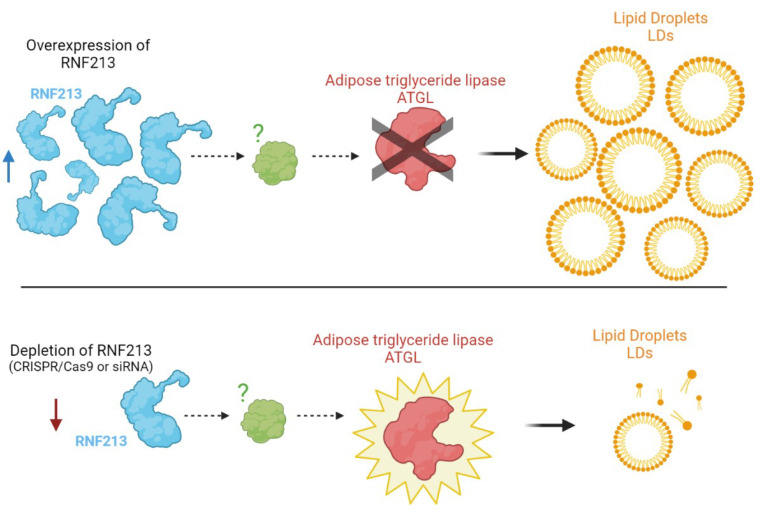
A possible link between RNF213 and lipid droplet (LD) stability. An overexpression of RNF213 increases the number and the covered area of LDs, while RNF213 depletion significantly reduced the abundance of LDs (siRNA, small interfering RNA; ATGL, adipose triglyceride lipase).

**Figure 3 ijms-23-04492-f003:**
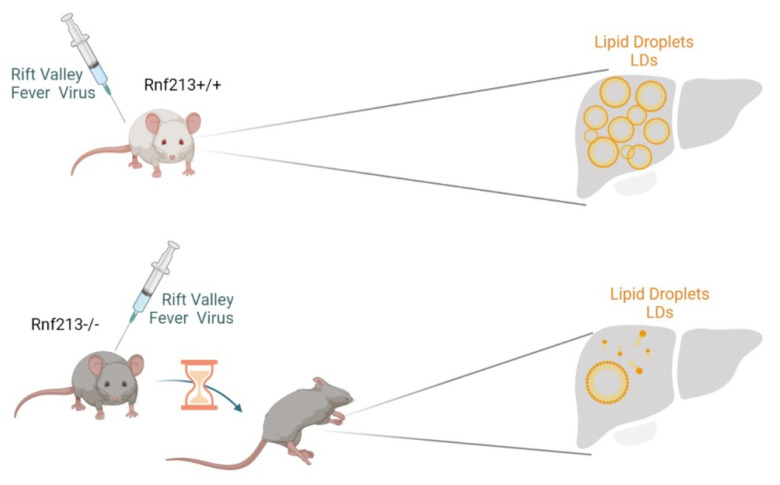
RNF213 protects against RVFV (Rift Valley Fever Virus) infection by increasing the level of LDs (lipid droplets).

**Figure 4 ijms-23-04492-f004:**
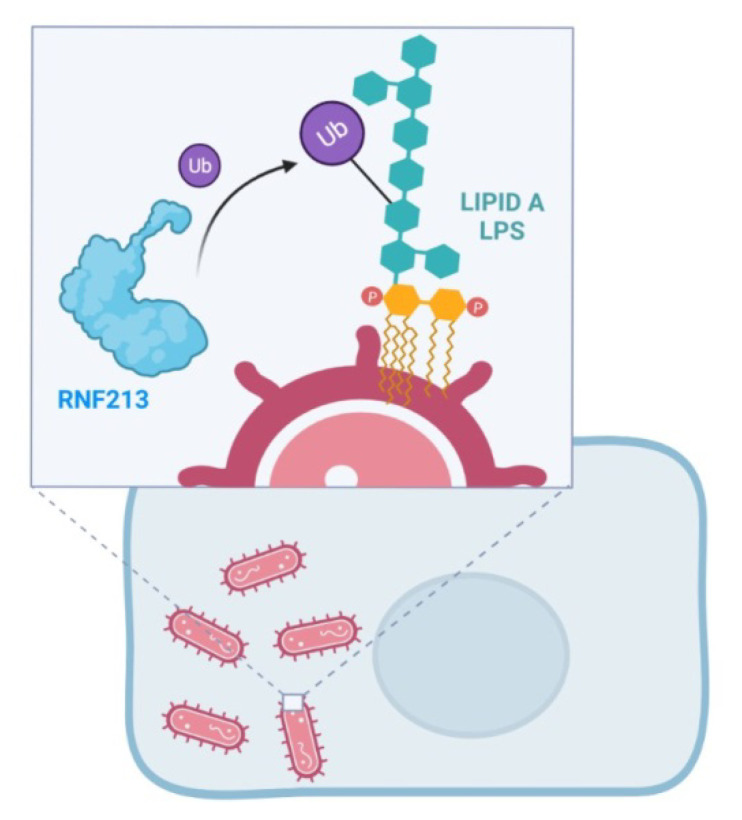
The RNF213 ubiquitin (Ub) E3 ligase domain mediates the ubiquitylation of Lipid A of bacterial lipopolysaccharide (LPS).

**Figure 5 ijms-23-04492-f005:**
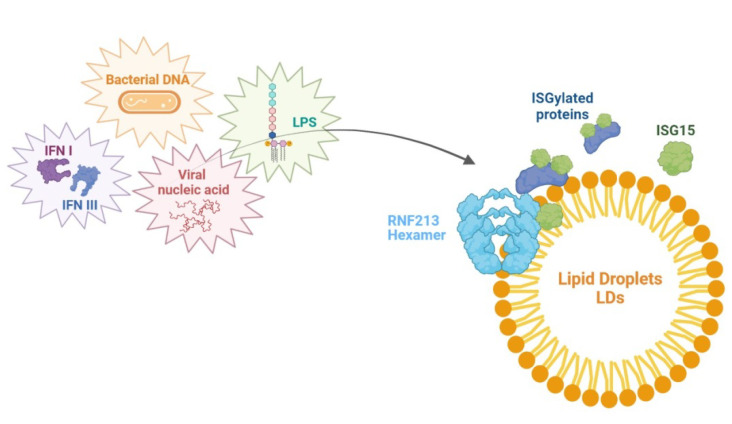
RNF213 oligomerization and translocation to lipid droplets (LDs) may serve for the creation of a binding platform for *Interferon-Stimulated Gene 15 (*ISG15) and multiple ISGylated proteins (IFN I, interferon I; IFN II, interferon II).

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
