# Peer review of "Novel Multifaceted Roles for RNF213 Protein"

_ijms, 2022, doi:10.3390/ijms23094492_

Round 1
Reviewer 1 Report
This review article seems to be redundant and appears to be just as the list of articles related to RNF213.
1.I think it is necessary to summarize more concisely as a whole.
2.It is not clear how each RNF213 function is related to moyamoya disease, what are the key molecules to disentangle the moyamoya disease pathology in Westerners who do not have RNF213 mutations; those are the authors’ own original questions. I recommend modifications in this regard.
3.It should be noted that mutations in RNF213 are not a single cause of moyamoya disease and that mutations in RNF213 are also found in diseases other than moyamoya disease.
Author Response
Answers to Reviewers
Reviewer 1
This review article seems to be redundant and appears to be just as the list of articles related to RNF213.
We are grateful to the Reviewer for his/her comments and for the careful revision. We are confident that the new information and conspicuos requested changes that we have introduced, will be convincing and will fill the gaps highlighted by the Referee.
1.I think it is necessary to summarize more concisely as a whole.
We have removed a substantial part of the text to streamline the manuscript and to avoid the redundancy highlighted by the Reviewer. Specifically, the word count (excluding the References), dropped to 5084 words from the previous 5822 words, also considering the novel sections included as requested by both Reviewers. The reductions pertained the whole manuscript, to simplify the comprehension of the text and to endorse the explanation and interpretation of novel RNF213 roles, as against the mere compilation of experimental details. All the changes -and especially the cuts in the text- being disseminated throughout the whole manuscript, are highlighted in the tracked changes version attached to the Revision. The list of References has been modified accordingly.
2.It is not clear how each RNF213 function is related to moyamoya disease, what are the key molecules to disentangle the moyamoya disease pathology in Westerners who do not have RNF213 mutations; those are the authors’ own original questions. I recommend modifications in this regard.
We thank the Referee for this comment that evidenced a limit of our work and pointed out the more general and still open question about the pathogenesis of such a rare and still obscure cerebrovascular condition. We have introduced a more “Westerners oriented” point of view of Moyamoya condition. Specifically, we have inserted this part at the end of paragraph 3 that has been re-nominated as “RNF213 genetic background”.
3.It should be noted that mutations in RNF213 are not a single cause of moyamoya disease and that mutations in RNF213 are also found in diseases other than moyamoya disease.
We agree with the Reviewer that RNF213 polymorfism/variants found in diseases other than Moyamoya arteriopathy are noteworthy, and also considering our purposes to provide an overall view of this peculiar and mysterious protein we better detailed such an issue. As rightly requested by the Reviewer, we specifically reported such evidence in the Conclusion.
Reviewer 2 Report
Thank you for the opportunity to review this manuscript. This is a thorough review describing the structure and role of RNF3 protein and it actions of development of Moya Moya disease. The authors have discussed the nuances of its role in lipid metabolism, inflammation and effects on micro-biome. It may be prudent to add a few lines about the pathological finding of fibrosis and smooth muscle cells which is seen in both hereditary and acquired Moya Moya which suggest the strong link between inflammation and RNF3 in these disorders.
Overall, this is a well written manuscript but I am unable to grasp the significance of the title and how it is pertinent to this manuscript. I do not share the author's sense of whimsy in academic writing and would recommend considering a more formal tone for the title. This title seems to distract from the serious work that the authors are trying to highlight.
Author Response
Answers to Reviewers
Reviewer 2
Thank you for the opportunity to review this manuscript. This is a thorough review describing the structure and role of RNF3 protein and it actions of development of Moya Moya disease. The authors have discussed the nuances of its role in lipid metabolism, inflammation and effects on micro-biome.
We are very grateful to the Reviewer for her/his comments and kind appreciation for our study.
It may be prudent to add a few lines about the pathological finding of fibrosis and smooth muscle cells which is seen in both hereditary and acquired Moya Moya which suggest the strong link between inflammation and RNF3 in these disorders.
We thank the Reviewer for this very useful suggestion. We have now introduced a few lines about the pathological finding of fibrosis and vascular smooth muscle cells in Moyamoya in paragraph 4, with a more understandable reference to the connection between inflammation and RNF213 functions in Moyamoya arteriopathy. The list of References has been modified accordingly.
Overall, this is a well written manuscript but I am unable to grasp the significance of the title and how it is pertinent to this manuscript. I do not share the author's sense of whimsy in academic writing and would recommend considering a more formal tone for the title. This title seems to distract from the serious work that the authors are trying to highlight.
We apologize for such a coarse and thoughtless use of an excessive informal tone for the manuscript Title. To remedy to this unconcern and to the lack of depth in the previous Title, we have currently removed the first part of the sentence including the wordplay, hoping to meet the approval of the Reviewer. The present Title is:
“Novel multifaceted roles for RNF213 protein”
We hope that the Reviewer agrees with the interpretation of her/his comment.